# Disaggregating the evidence linking biodiversity and ecosystem services

Taylor H. Ricketts[1,2], Keri B. Watson[1,2], Insu Koh[1,2], Alicia M. Ellis[1,3], Charles C. Nicholson[1,2], Stephen Posner[1,2], Leif L. Richardson[1,2] & Laura J. Sonter[1,2]

Ecosystem services (ES) are an increasingly popular policy framework for connecting biodiversity with human well-being. These efforts typically assume that biodiversity and ES covary, but the relationship between them remains remarkably unclear. Here we analyse >500 recent papers and show that reported relationships differ among ES, methods of measuring biodiversity and ES, and three different approaches to linking them (spatial correlations, management comparisons and functional experiments). For spatial correlations, biodiversity relates more strongly to measures of ES supply than to resulting human benefits. For management comparisons, biodiversity of 'service providers' predicts ES more often than biodiversity of functionally unrelated taxa, but the opposite is true for spatial correlations. Functional experiments occur at smaller spatial scales than management and spatial studies, which show contrasting responses to scale. Our results illuminate the varying dynamics relating biodiversity to ES, and show the importance of matching management efforts to the most relevant scientific evidence.

[1] Gund Institute for Ecological Economics, University of Vermont, 617 Main Street, Burlington, Vermont 05405, USA. [2] Rubenstein School of Environment and Natural Resources, University of Vermont, 81 Carrigan Drive, Burlington, Vermont 05405, USA. [3] Pathology & Laboratory Medicine, College of Medicine, University of Vermont, 89 Beaumont Avenue, Burlington, Vermont 05405, USA. Correspondence and requests for materials should be addressed to T.H.R. (email: taylor.ricketts@uvm.edu).

Improving human well-being without destroying our planet's natural resources is a central challenge of the twenty-first century[1–5]. Ecosystem services (ES) can help align these goals by clarifying the benefits natural systems provide to people[6,7]. In seeking mutual benefits for society and biodiversity, governments, non-profit organizations and international institutions increasingly incorporate ES into planning and decisions[8–11].

These efforts depend on the assumption that biodiversity and ES covary[12–14], but links between them remain remarkably unclear[15–18]. Relatively few studies test the relationship empirically, and those that do show inconsistent results[17,18]. Studies also are scattered across diverse disciplines, vary in their analytical approaches, measure both biodiversity and ES in differing ways, occur at a wide range of scales and often focus on specific management issues instead of general scientific hypotheses[15,17]. Finally, dynamics linking biodiversity and ES vary; some ES depend more on overall abundance or biomass, while others depend on particular functional groups or key species[17,19].

Previous syntheses on this topic have typically pooled studies across these and other factors, reporting general findings that may mask important structure in the evidence[16–18]. In comparison, the literature linking biodiversity and basic ecosystem functions (for example, primary productivity and nutrient cycling) is well developed, and syntheses have shown generally significant and positive relationships[12,18,20,21]. Equivalent syntheses for the relationship between biodiversity and ES, especially of studies involving real-world scales and conditions, have been lacking.

Here we aim to more deeply understand the evidence linking biodiversity and four services: carbon storage; pest control; crop pollination; and water purification. We assess more than 500 recent peer-reviewed papers and quantify the distribution of positive, negative and non-significant relationships (hereafter, the 'balance of evidence'). We then compare that balance of evidence among five factors that may influence relationships between biodiversity and ES.

First, we identify three recurring approaches for linking biodiversity and ES (hereafter linkage types): 'spatial linkages', which compare levels of biodiversity and ES across space; 'management linkages', which compare responses of biodiversity and ES to the same management intervention; and 'functional linkages', which test specifically whether ES are a mechanistic function of biodiversity. (Fig. 1; see results for more detail on linkage types.)

Second, we distinguish studies that measure ES as either biophysical supply or as benefits of this supply to people[9,22,23]. For example, crop pollination can be measured as wild bee visits to crop flowers (supply) or as improved crop production (benefit). Supply and benefit measures are not necessarily correlated; social and economic factors also affect benefits and can weaken the signal of biodiversity on ES[23]. For example, diverse landscapes can increase visits by wild pollinators but not improve crop production[24], perhaps because farmers keep honey bees or grow self-fertile varieties.

Third, we distinguish studies that link ES to the diversity of expected 'service providers'[25], as opposed to communities or functional groups that are mechanistically unrelated to the ES of interest. For example, an ecologist testing diversity theory may examine spatial relationships between carbon storage and plant diversity (service provider)[26], while a conservationist siting protected areas might focus on whether carbon storage covaries spatially with diversity of birds (non-service provider)[27].

Fourth, we distinguish studies that relate ES to biodiversity at different levels of organization, from genes to ecosystems[17,19]. Species are relatively clear units of both evolution and management, but higher-order levels of organization can also underpin ES supply and benefit. For example, crop pollination and carbon storage have been shown to be more strongly related to functional group diversity than to species diversity[28,29].

Finally, we examine the influence of spatial extent (that is, overall area studied) and grain (that is, area of each sample unit) on the balance of evidence for each linkage type. Spatial scale is widely thought to influence the links between biodiversity and ES[19], but proposed mechanisms vary and empirical tests are few[15,30].

Overall, our synthesis shows that reported relationships differ markedly among ES, ways of measuring biodiversity and ES, and approaches to relating them. Disaggregating the evidence in this way illuminates the varying dynamics linking biodiversity and ES, and illustrates the importance of matching management efforts to the most relevant scientific evidence.

## Results

**Quantity of evidence.** We read at least the top-ranked 100 papers published between 2001 and 2014 for each ES, according to Web of Science (Table 1; Methods). Only 14% of these papers met our two criteria for inclusion: they quantified both biodiversity and at least one ES; and they statistically related them in some way (Table 1). Studies occurred in 30 countries but were concentrated in North America and Europe (Supplementary Fig. 1). Papers often reported multiple relationships between biodiversity and ES (mean: 2.3), yielding 186 relationships overall (Table 1). For clarity we refer to statistically significant relationships as simply 'positive' or 'negative'.

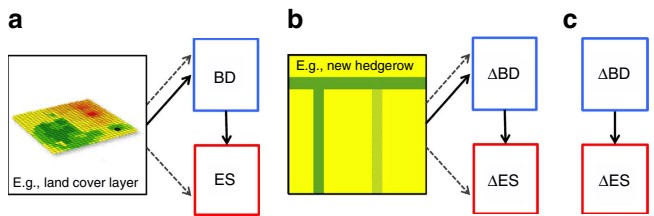

**Figure 1 | Three types of linkage between biodiversity (BD) and ES.** (**a**) Spatial linkage, (**b**) management linkage and (**c**) functional linkage. For **a**,**b**, BD and ES could be linked causally (solid arrows; biodiversity of 'service providers') or respond in parallel to spatial or management factors (dashed arrows; biodiversity of 'non-service providers').

**Table 1 | Numbers of papers and relationships included in the study.**

| Service | Papers read | Papers included | Relationships | Included/read | Relationships/included |
|---|---|---|---|---|---|
| Carbon storage | 96 | 28 | 62 | 0.29 | 2.21 |
| Crop pollination | 185 | 25 | 55 | 0.14 | 2.20 |
| Pest control | 107 | 23 | 54 | 0.21 | 2.35 |
| Water purification | 197 | 5 | 15 | 0.03 | 3.00 |
| All ecosystem services | 585 | 81 | 186 | 0.14 | 2.30 |

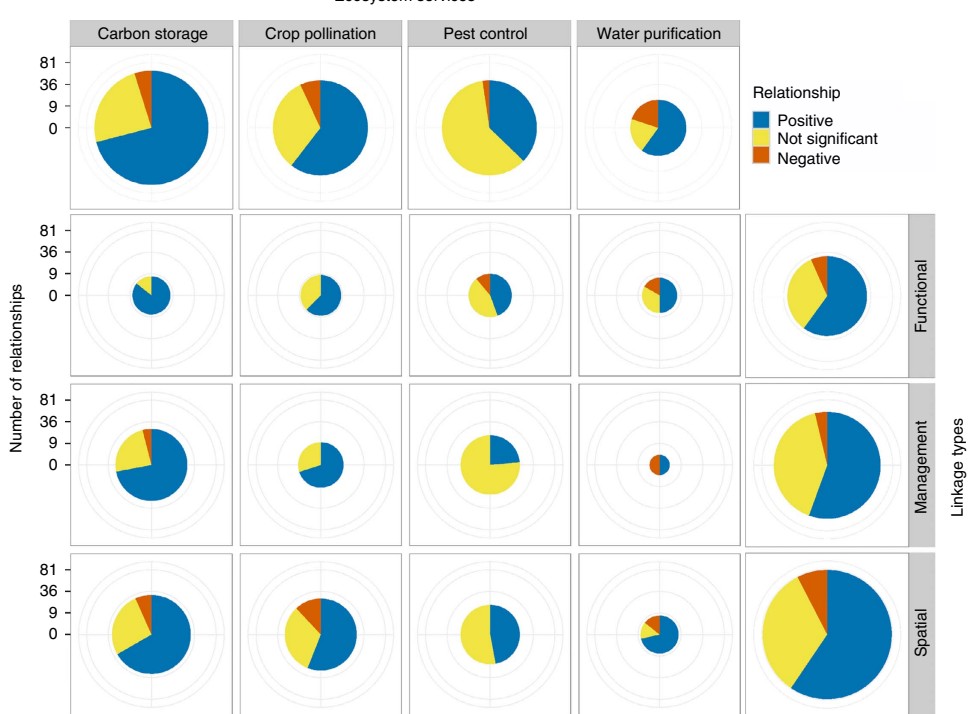

**Figure 2 | Evidence for linkages between biodiversity and ES.** Top row: all relationships for each of four ES reviewed (that is, pooling all linkage types; G-test, G = 19.96, degrees of freedom (d.f.) = 6, P = 0.003, n = 163). Right column: all relationships for each linkage type (that is, pooling all ES; G-test, G = 1.57, d.f. = 4, P = 0.814, n = 163). Pie charts are scaled such that their areas (not radii) are proportional to the number of relationships. Colours depict the sign of the reported relationship: blue = positive; red = negative; yellow = not significant. See Fig. 1 for description of three linkage types, and see Supplementary Table 1 for data in tabular form.

**Three approaches to linking biodiversity and ES**. From these papers, we identified three recurring approaches to evaluating links between biodiversity and ES. We refer to these as linkage types. Each linkage type involves differing biophysical and social phenomena, and each has different strengths and weaknesses regarding the scientific inferences they support. We describe linkage types here and compare balance of evidence among them and other factors in the following section.

For spatial linkages (Fig. 1a), researchers measure levels of biodiversity and ES at several sites across a landscape or region, and then compare those patterns statistically. The two variables could be linked mechanistically or respond similarly to other spatial factors. Inference for a mechanistic relationship is only appropriate if biodiversity of likely service providers[25] is reported. Variables are either estimated with spatially explicit models or observed directly from field measurements. For example, Zhang et al.[26] sampled plant communities and carbon storage across a landscape in southwestern China. They found a negative relationship between the plant diversity and aboveground biomass, but no relationship with soil organic carbon.

For management linkages (Fig. 1b), researchers measure the response of both biodiversity and ES to a difference in management or land use. Comparisons can be through time in a particular place, or among sites at the same time. As with spatial linkages, the two variables could be linked mechanistically or respond similarly to other factors, but mechanistic inference is appropriate only if relationships involve service providers. For example, Morandin et al.[31] compared predator diversity and pest abundance in tomato fields with and without hedgerows. They found higher predator diversity in hedgerows, and greater predator abundance and lower pest abundance in adjacent fields.

For functional linkages (Fig. 1c), researchers manipulate levels of biodiversity in a lab or field experiment, and measure the response of ES. The goal is explicitly to test the functional relationship between biodiversity and ES; experimental designs are used to isolate that relationship, which always involves service providers. For example, Cardinale[32] manipulated the diversity of algae in microcosms (from 1 to 8 species) and showed that more diverse communities remove more nitrogen from water than do less diverse communities under heterogeneous stream flow conditions.

**Balance of evidence**. We find the balance of evidence differs significantly among ES (Fig. 2, top row). Only 37% of relationships are positive for pest control, while 60–71% are positive for the other three ES. Carbon storage, crop pollination and pest control include small fractions of negative relationships, while 20% of relationships for water purification are negative.

The balance of evidence is remarkably similar among linkage types when all four ES are pooled (Fig. 2, right-hand column). Each ES, however, displays a distinct pattern in balance of evidence among linkage types. With carbon storage, for example, the frequency of positive relationships increases from spatial (67%) to management (72%) to functional (86%) linkages, perhaps reflecting an increasing degree of experimental control[7,9]. Other ES display contrasting patterns (Fig. 2).

ES differ in the frequency of relationships reporting ES supply or benefit (Fig. 3a). Because social and economic factors can weaken the signal of biodiversity on benefits[23] (see Introduction), we expected significant relationships to be less frequent for benefit compared with supply. Pooling relationships, we find no such differences (Fig. 3b). Examining linkage types individually, however, we find that spatial linkages show the expected pattern: non-significant relationships are more common for benefit than for supply (Fig. 3b). We also examined the few papers that report

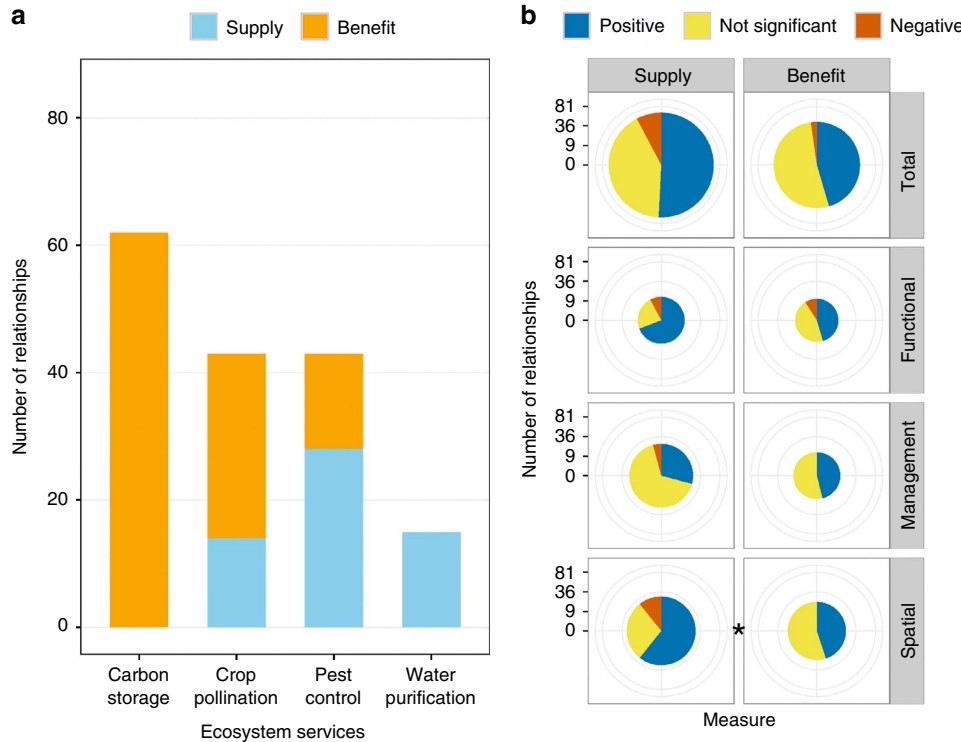

**Figure 3 | Differences in evidence between supply and benefit.** ES are measured either as biophysical supply (for example, pollinator abundance) or as realized benefits to people (for example, increases in crop yield). (**a**) Number of relationships reporting supply or benefit. Blue denotes ES supply and orange denotes ES benefit. No water purification studies measured benefits, and carbon storage provides a global benefit, so measures of supply (for example, C stored) are also measures of benefit (for example, climate forcing avoided). If relationships for both supply and benefit were reported in the same paper, only the benefit relationship is included in this panel. (**b**) Comparing the balance of evidence between studies of supply and benefit (G-tests; total: $G = 2.38$, degrees of freedom (d.f.) $= 2$, $P = 0.304$, $n = 109$; functional: $G = 1.50$, d.f. $= 2$, $P = 0.473$, $n = 24$; management: $G = 1.76$, d.f. $= 2$, $P = 0.415$, $n = 37$; spatial: $G = 5.80$, d.f. $= 2$, $P = 0.055$, $n = 48$). (**b**) Includes supply and benefit relationships from the same study if both were reported, but includes only pest control and crop pollination services—those in which both ES measures were reported (**a**). Pie charts are scaled such that their areas (not radii) are proportional to the number of relationships. Colours depict the sign of the reported relationship: blue = positive; red = negative; yellow = not significant. See Fig. 1 for description of three linkage types. Labels for significance: *$P < 0.10$.

**Table 2 | Results from studies reporting relationships with both ES supply and ES benefit.**

|  | Benefit | | |
|---|---|---|---|
|  | **Positive** | **Not significant** | **Negative** |
| *Supply* |  |  |  |
| Positive | 8 | 3 |  |
| Not significant | 1 | 9 |  |
| Negative |  | 1 | 1 |

biodiversity relationships with both ES supply and benefit. Consistent with our expectation, of 13 papers with a significant relationship for supply, 4 (31%) report a non-significant relationship with benefit (Table 2).

While a majority of the overall evidence involves service providers, more than half of relationships for water purification, and a substantial minority for the other three ES, do not (Fig. 4a). Because of their mechanistic roles, we expected service providers to be positively related to ES more frequently than non-service providers. For management linkages, the evidence supports this expectation (Fig. 4b), but spatial linkages show the opposite trend. As with previous results, pooling relationships across linkage types masks these differences (Fig. 4b).

Most studies measure biodiversity at the species level (Fig. 5a). However, the balance of evidence does not differ between these

studies and those using higher levels of organization (Fig. 5b), although the small number of studies in the latter category makes interpretation difficult.

As expected, functional linkages tend to be reported at finer scales than spatial or management linkages (Fig. 6). Scale also appears to affect the balance of evidence, but in contrasting ways. For management linkages, positive relationships tend to be reported at finer grains and extents (Fig. 6b), while for spatial linkages, positive relationships tend to be reported more often at large extents (Fig. 6c).

## Discussion

Taken together, our findings suggest that pooled evidence linking biodiversity and ES could mislead both scientific syntheses and management interventions. We repeatedly find important differences in the balance of evidence that are masked by pooled data. Accounting for these differences will help to deepen our theoretical understanding of the relationship between biodiversity and ES, and support management efforts with the most relevant scientific evidence.

Linkage types appear to interact with spatial scale, service providers and supply versus benefit measures in determining the relationship between biodiversity and ES. For spatial scale, we find opposing effects between management and spatial linkage types (Fig. 6). Within management linkages, fewer positive relationships at larger grains and extents may indicate that broad-scale studies incorporate more variability within and

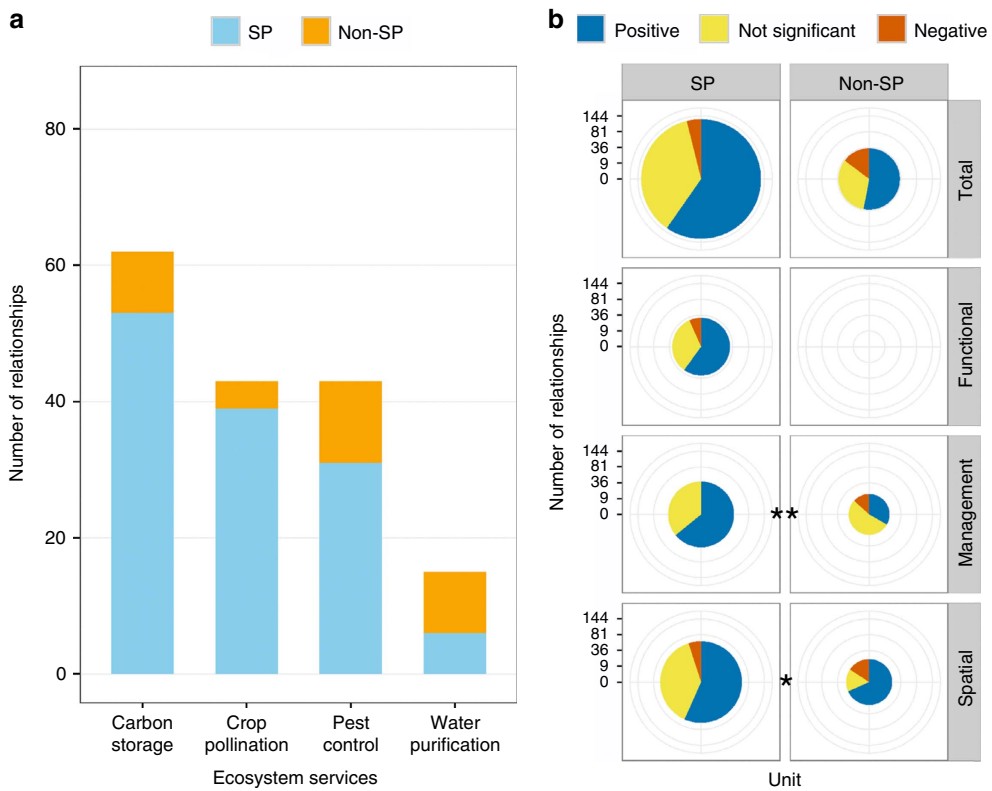

**Figure 4 | Differences in evidence between service providers and non-service providers.** Service providers are the taxa, communities or functional groups expected to be responsible for the ES of interest. (**a**) Number of relationships involving service providers (SP) and non-service providers (non-SP). Blue denotes service providers and orange denotes non-service providers. (**b**) Comparing the balance of evidence between service providers and non-service providers (G-tests; total: $G = 4.50$, degrees of freedom (d.f.) $= 2$, $P = 0.106$, $n = 163$; management: $G = 7.94$, d.f. $= 2$, $P = 0.019$, $n = 54$; spatial: $G = 4.82$, d.f. $= 2$, $P = 0.090$, $n = 79$). Pie charts are scaled such that their areas (not radii) are proportional to the number of relationships. Colours depict the sign of the reported relationship: blue $=$ positive; red $=$ negative; yellow $=$ not significant. See Fig. 1 for description of three linkage types. Labels for significance: $^*P < 0.10$; $^{**}P < 0.05$.

among replicates, masking any effects. Within spatial linkages, in contrast, there may be more positive relationships at large extents because broad-scale studies capture greater variation in factors that influence both biodiversity and ES. For service providers (Fig. 4), we again find opposite effects between management and spatial linkages. While management linkages support our expectation that service providers are more positively related to ES, the opposite was true for spatial linkages, perhaps because positive relationships with non-service providers tend to be investigated at large spatial scales. For supply versus benefit (Fig. 3), we found expected differences in the balance of evidence only for spatial linkages. However, only two of the four ES we analysed included both supply and benefit measures, and very few individual papers included both, making inference difficult.

Our findings underscore the caution required when applying evidence from one linkage type to decisions that relate to another (for example, assuming spatial correlations reflect functional links, or applying results from small-scale experiments to predict broad-scale spatial patterns[30,33,34]). Pooled evidence may also misinform decisions by blurring differences among linkage types. Instead, carefully matching specific policies or interventions with the most relevant evidence can strengthen efforts to enhance biodiversity and ES. For example, national policies to build efficient protected area networks should draw on evidence from spatial linkages (for example, refs 10,27,35), while local efforts to restore hedgerows within agricultural landscapes require evidence from management linkages (for example, refs 14,31).

The four 'regulating' services we examine are of course a limited sample of all ES provided by the world's ecosystems[2]. We selected them because they are of wide scientific and policy interest, involve differing ecological, spatial and social dynamics, draw from widely divergent disciplines and literatures, and are described by a sufficient number of publications[18,36,37]. Despite our small sample, we find important differences in the balance of evidence among ES (Fig. 2). Understanding whether the patterns we report here represent those for other ES will require extending our approach to a wider sample of ES[2]. We would expect that other regulating ES (for example, soil processes) are likely to increase with the diversity of relevant species and functional groups[18]. Provisioning ES (for example, timber production) typically depend on the abundance of harvested species, which may or may not increase with biodiversity[18]. Cultural ES (for example, spiritual values) may relate as much to specific species or features in the landscape as to biodiversity *per se*[38]. For many ES, literature on the effects of biodiversity is not yet sufficient to support synthesis[16,17].

Our findings point to several important directions for future research. First, we identify poorly documented linkages (for example, management studies of water purification; Fig. 2), where additional research would most powerfully advance our understanding of the conditions under which biodiversity and ES are related. Second, the role biodiversity plays in conferring actual human benefits is understudied[9,39,40], especially for certain ES (Fig. 3). While measuring supply is often more straightforward than quantifying benefits, the latter are the unique features of ES

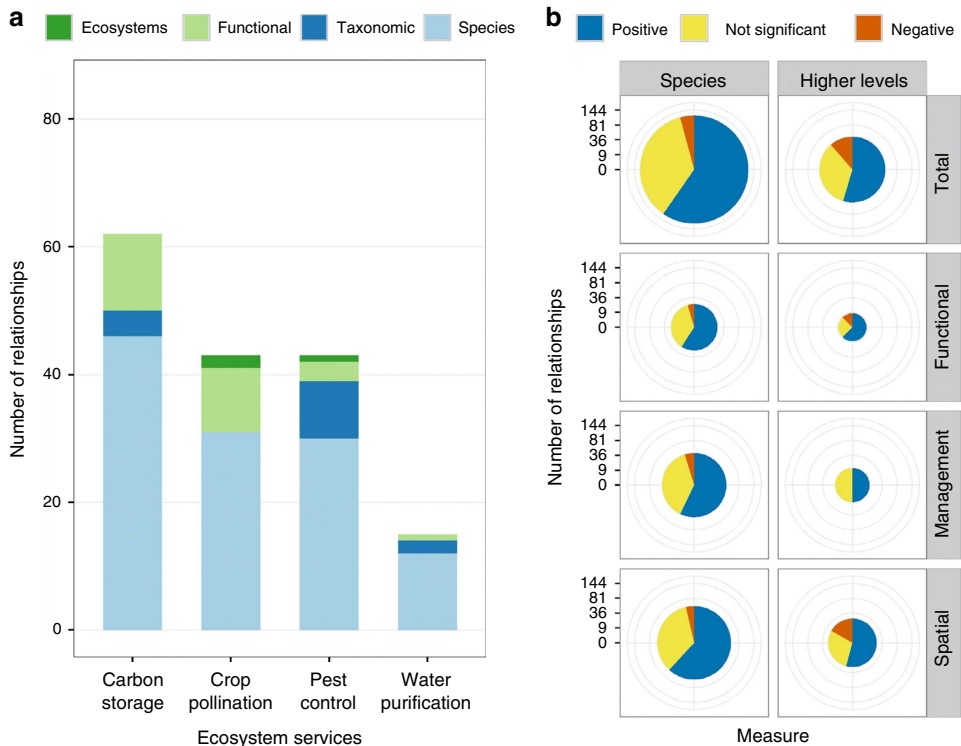

**Figure 5 | Differences in evidence among levels of biodiversity.** Biodiversity was measured as variety at one of four levels: species; functional groups; taxonomic groups broader than species; and ecosystems. (**a**) Number of relationships reporting biodiversity at each level. Colours denote levels of biodiversity measured: green = ecosystems; light green = functional groups; blue = taxonomic groups; light blue = species. (**b**) Balance of evidence for studies measuring species diversity, compared with studies measuring diversity at higher levels of organization (G-tests; total: $G = 2.56$, d.f. = 2, $P = 0.278$, $n = 163$; functional: $G = 0.74$, d.f. = 2, $P = 0.689$, $n = 30$; management: $G = 1.40$, d.f. = 2, $P = 0.496$, $n = 54$; spatial: $G = 3.66$, d.f. = 2, $P = 0.161$, $n = 79$). Pie charts are scaled such that their areas (not radii) are proportional to the number of relationships. Colours denote the sign of the reported relationship: blue = positive; red = negative; yellow = not significant. See Fig. 1 for description of three linkage types.

that make them both scientifically novel and policy relevant[9,10,39]. Third, more evidence is needed on the role of biodiversity at multiple levels of organization (Fig. 5)[15,18]. While most studies continue to focus on species, biodiversity at higher (for example, functional group) and lower (for example, population) levels of organization can also underpin ES supply and benefit[28,29,41]. Fourth, we need to better characterize the shape of relationships between biodiversity and ES. Both theory and experiments suggest the possibility of saturating or other non-linear effects[19,29,42,43], but in our sample of 186 relationships only 6 report non-linear tests. Finally, there is clear value in understanding the relationship between ES and both service providers and non-service providers (Fig. 4). The former can complement experiments in examining the functional role of biodiversity[15,16], while the latter can inform management interventions that increasingly aim to benefit both nature and people[10,27].

Building from our synthesis to develop a broad registry of evidence[44] would help scientists, managers and policy-makers match potential interventions with the most relevant scientific information. Such a registry would particularly strengthen ongoing global efforts such as the Intergovernmental Platform on Biodiversity and Ecosystem Services[45], which aims to assess existing knowledge for national governments. More broadly, it would help decision-makers with the essential task of sustaining nature and human well-being together in a rapidly changing world.

## Methods

**Literature search.** We used ISI Web of Knowledge to search for articles published during 2001–2014, using the same search terms as Cardinale et al.[18]

(Supplementary Table 2). We initially reviewed the top 100 papers for each ES, as ranked by ISI relevance. We omitted reviews and meta-analyses to focus on primary studies and to allow coding of attributes not reported in syntheses. We included only papers that relate some measure of ES to a measure of biodiversity, using experimental, observational or modelled data to test for statistical significance. After reading 100 papers, if we had not accumulated at least 50 relationships for a given ES (see below), we continued reading, stopping when we either (i) surpassed 50 coded relationships or (ii) reached the 200th ranked paper. Overall, we read 585 papers and included 81 (Table 1).

**Coding relationships.** For each paper, we coded specific relationships reported between biodiversity and ES. One paper may be the source of several relationships, and we coded each relationship separately and treated them as independent observations. We categorized relationships as positive, negative or non-significant, based on authors' statistical results and using a threshold of $P = 0.05$. Only 6 of 186 relationships reported non-linear effects, so we recorded only the overall sign of relationships. Positive relationships occurred where biodiversity and ES were either both improved, or both diminished. In cases where ES was measured as reduction of some negative outcome (for example, decreased crop damage due to increased predator diversity; Supplementary Table 3), we reversed the sign of the relationship so that positive changes always represent desired outcomes.

For numerous studies, authors tested the influence of spatial or management variables on biodiversity and ES separately, but did not relate the two variables directly. In these cases, we inferred a biodiversity–ES relationship from the sign and significance of the separate results.

For each relationship we also recorded the following attributes: (i) type of linkage (for example, spatial, management and functional; Fig. 1); (ii) the level of organization at which biodiversity was considered (that is, genetic, species, taxonomic, functional or ecosystem); (iii) the metric used to quantify biodiversity (that is, richness, diversity index or abundance); (iv) whether or not biodiversity measures focused on taxa expected to provide the ES of interest (that is, 'service providers'[25]); (v) the ES outcome measured (that is, biophysical supply such as pollinator abundance, or realized benefits such as increased crop yield; Supplementary Table 3); (vi) spatial scale at which the research occurred (that is, grain and extent—either reported from paper or inferred from methods information if not reported quantitatively); (vii) whether the reported

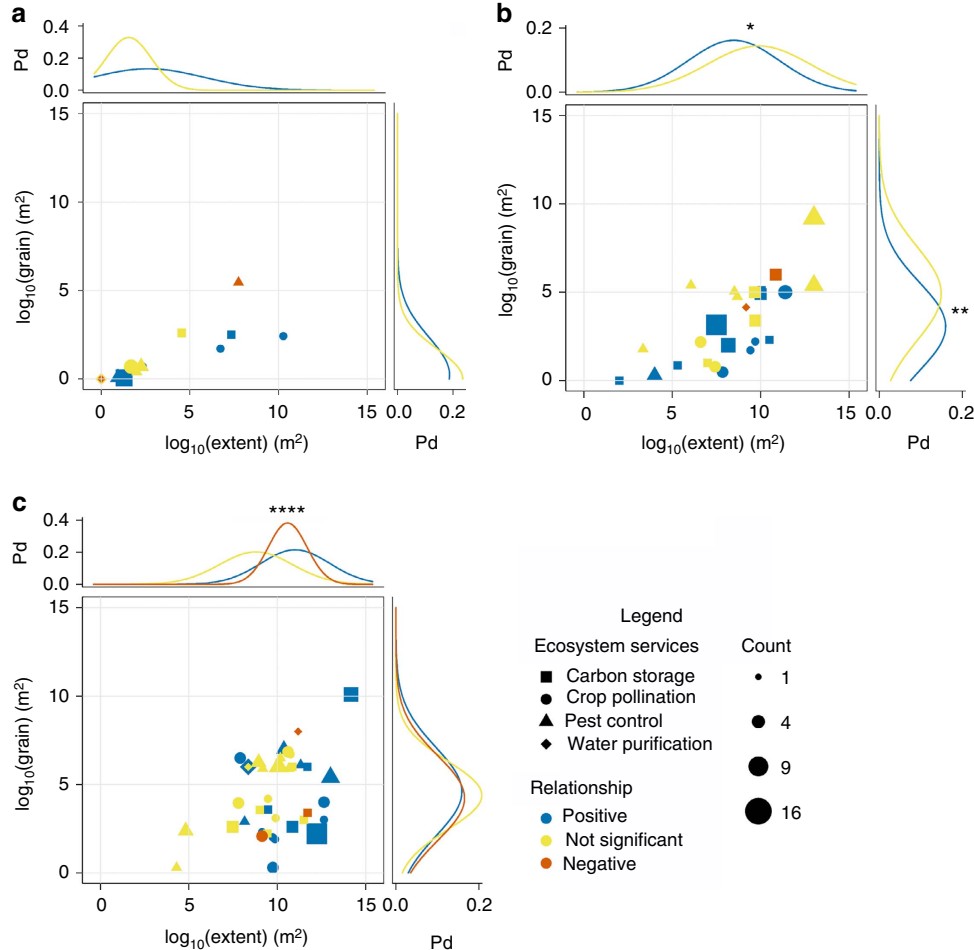

**Figure 6 | Effects of spatial scale on biodiversity–ES relationships.** (**a**) Functional linkages, (**b**) management linkages, (**c**) spatial linkages. Extent (*x* axes) is the overall area considered in the study, while grain (*y* axes) is the area of each sample unit. Symbols represent individual reported relationships. Symbol shapes represent ES: square = carbon storage; circle = crop pollination; triangle = pest control; diamond = water purification. Colours of symbols and curves depict the sign of the reported relationship: blue = positive; red = negative; yellow = not significant. Symbol sizes depict the number of relationships that share the same grain, extent and sign (typically relationships from the same study). Curves above and to the right of each panel depict Gaussian probability distributions (Pd) for positive, non-significant and negative relationships. Red curves are omitted in **a,b** due to few negative relationships. Analysis of variance results for differences in scale among positive, non-significant and negative relationships: **a**, extent: $F_{1,26} = 1.05$, $P = 0.316$, $n = 28$; **a**, grain: $F_{1,26} = 0.35$, $P = 0.56$, $n = 28$; **b**, extent: $F_{1,48} = 3.14$, $P = 0.083$, $n = 50$; **b**, grain: $F_{1,48} = 5.94$, $P = 0.019$, $n = 50$; **c**, extent: $F_{2,68} = 9.56$, $P < 0.001$, $n = 71$; **c**, grain: $F_{2,68} = 0.09$, $P = 0.907$, $n = 71$. Labels for significance: *$P < 0.10$; **$P < 0.05$; ****$P < 0.001$.

relationship was linear or non-linear; and (viii) country in which the study took place.

To code these attributes consistently and clearly, we developed a detailed set of decision rules (Supplementary Note 1). We improved inter-rater reliability by initially coding five papers independently, comparing results and clarifying the decision rules and thresholds. All included papers were then coded independently by two different authors, with each person coding some papers from all four ES. Discrepancies were discussed and resolved by all coding authors, with refinements to decision rules made as necessary.

The variables reported in this paper are a subset of those we coded. Owing to sample size and space constraints, we occasionally pooled relationships across important variables in our analyses. For example, we record the taxa involved in each study (Supplementary Note 1), but our analyses pool across taxa to focus on service providers and levels of organization. Many additional analyses are likely possible. In addition, we compile reported relationships regardless of whether authors attempt to account for confounding effects, because deciding whether confounders have been adequately controlled is difficult to do consistently.

**Analyses.** We analysed the distribution of relationships among positive, negative and non-significant categories, and refer to this distribution as the 'balance of evidence'. We compare the balance of evidence among ES, linkage types, measures of ES and biodiversity, and spatial scales. This simple 'vote-counting' approach, while less powerful than formal meta-analysis, allowed us to code studies consistently across widely varying disciplines, using data typically reported in the studies themselves[46]. We tested differences in the distribution of relationships

using likelihood-ratio tests (G-test), and tested differences in scale among positive, negative and non-significant relationships using analysis of variance. We used R for all analyses[47], including the 'likelihood.test{Deducer}[48] and 'Anova{car}' libraries[49]. To consistently present findings despite occasional low sample size and power, we interpreted any result with $P < 0.10$ as significant, but we mark levels of significance clearly throughout.

Some papers ($n = 23$) related the same biodiversity measure to both ES supply and benefit. To avoid double counting, we include only the benefit relationships from these papers, except in Fig. 3b, which explicitly compares supply and benefit measures. Some authors defined biodiversity as abundance, instead of as the variability among types at genetic, species or higher levels. To focus on the linkages between ES and biodiversity *per se*, we analysed only relationships that report some measure of diversity. Table 1 reflects this final stage of omissions.

As in any literature synthesis, our findings may be sensitive to the search terms and search tool used to identify relevant papers. We explore these sensitivities in Supplementary Methods and Supplementary Fig. 2. Our findings also likely include reporting bias, in which non-significant or otherwise less-compelling results are less likely to be published[50]. The strength of reporting bias may differ among the widely divergent fields contributing relevant literature to the study of ES.

**Data availability.** The publication and attribute coding data that support the findings of this study are available in figshare (www.figshare.com) with DOI: 10.6084/m9.figshare.3775821 (ref. 51). The protocol used to code papers is described in Supplementary Note 1.

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

## Acknowledgements

We thank B. Cardinale, K. Carney, B. Chaplin-Kramer, A. Claussen, B. Fisher, R. Naidoo, N. Sanders and the Gund Institute community for comments. We are grateful for support from the Lintilhac Foundation and from the Gund and Parker families. I.K. is supported by USDA-NIFA award #2012-51181-20105. C.C.N. is supported by an NSF Graduate Research Fellowship under Grant #DGE-1451866. L.L.R. is supported by USDA NIFA award #2015-67012-22770. L.J.S. and K.B.W. are supported by USDA McIntire-Stennis award #2014-32100-06050 to the University of Vermont.

## Author contributions

T.H.R. conceived and oversaw the project; all authors contributed to study design and collected data; I.K., L.J.S. and K.B.W. performed analyses; T.H.R. and K.B.W. wrote the manuscript. All authors discussed results and implications and edited the manuscript at all stages.

## Additional information

**Competing financial interests:** The authors declare no competing financial interests.

