## [Peer Review File · Nature Communications]

PEER REVIEW FILE

Reviewers' Comments:

Reviewer #1 (Remarks to the Author)

This manuscript is a revised version of a previous submission concerning the relationship between biodiversity and ecosystem functions and or services. The approach is a structured literature review; a meta-analysis of sorts. Its findings are that the relationship, though more or less positive, is sensitive to issues such as scale, the particular service, and the nature of the study (i.e., was it experimental or observational). Its conclusions do not bear on the science, per se, but it does provide a synopsis of the state of the body of research (as sampled and structured by the authors methodology) and cautions about matching management objectives with properties of the research it references.

The methods, validity of approach, statistical inference (though limited), and quality of the presentation are quite good. Indeed, this version is markedly improved over the last.

The paper is well written, its conclusions more robust than in its previous iteration.

I have no particular comments to add since I provided substantial commentary for the first revision and the authors have addressed all of them as best they can. Certain issues remain, such as the somewhat unclear motivation, fuzzy implications, and masked limitations (and strengths) of scientific research that the coding process engenders. I don't think managers, for example, benefit from this – they want clear and plain guidelines, whereas this is a cautionary note that requires managers to figure out how best to use the information. Also, the field evolves. But every study of this nature will find no way around these problems and I think the authors have done a genuine and sincere effort to present their case in the most robust fashion they can.

I strongly support the study. It is an important contribution to the field.

Response to Reviewers

We appreciate the final review comments from Reviewer #1. We've pasted those comments below, with very brief responses in all caps.

Reviewer #1 (Remarks to the Author):

This manuscript is a revised version of a previous submission concerning the relationship between biodiversity and ecosystem functions and or services. The approach is a structured literature review; a meta-analysis of sorts. Its findings are that the relationship, though more or less positive, is sensitive to issues such as scale, the particular service, and the nature of the study (i.e., was it experimental or observational). Its conclusions do not bear on the science, per se, but it does provide a synopsis of the state of the body of research (as sampled and structured by the authors methodology) and cautions about matching management objectives with properties of the research it references.

The methods, validity of approach, statistical inference (though limited), and quality of the presentation are quite good. Indeed, this version is markedly improved over the last.

THANK YOU.

The paper is well written, its conclusions more robust than in its previous iteration.

AGAIN, THANKS AND GOOD TO HEAR.

I have no particular comments to add since I provided substantial commentary for the first revision and the authors have addressed all of them as best they can. Certain issues remain, such as the somewhat unclear motivation, fuzzy implications, and masked limitations (and strengths) of scientific research that the coding process engenders. I don't think managers, for example, benefit from this – they want clear and plain guidelines, whereas this is a cautionary note that requires managers to figure out how best to use the information. Also, the field evolves. But every study of this nature will find no way around these problems and I think the authors have done a genuine and sincere effort to present their case in the most robust fashion they can. I strongly support the study. It is an important contribution to the field.

WE AGREE WITH THE UNIVERSAL LIMITATIONS OF CONDUCTING A STUDY OF THIS KIND. AND WE APPRECIATE YOUR HELP IN MAKING OURS AS STRONG AS IT CAN BE.

Sincerely,

Taylor Ricketts